# Few-Mode Fiber Characterization System Based on the Spatially and Spectrally Imaging Technique [note 1]

**DOI:** 10.3390/s22051809

**Published:** 2022-02-25

**Authors:** Jianxun Yu, Fengze Tan, Changyuan Yu

**Affiliations:** 1Photonics Research Centre, Department of Electronic and Information Engineering, The Hong Kong Polytechnic University, Hong Kong, China; jianxyu@polyu.edu.hk; 2The Hong Kong Polytechnic University Shenzhen Research Institute, Shenzhen 518057, China; f.z.tan@connect.polyu.hk

**Keywords:** few-mode fiber, mode dispersion, mode coupling

## Abstract

With the widespread use of few-mode fibers, mode characteristics testing becomes essential. In this paper, current few-mode fiber testing techniques are discussed, and the S^2^ imaging technique is chosen and demonstrated to be capable of few-mode fiber characterization in principle. As a result, the few-mode fiber characterization system with the S^2^ imaging technique is built and used to obtain accurate mode dispersion of two-mode fibers (a commonly used few-mode fiber) of different lengths. Then, various filters are applied to extract the fundamental and high-order modes to acquire mode coupling components (discrete and distributed mode coupling). The proposed system spectrally characterizes the few-mode fiber by resolving the interference information from the superimposed optical field spatially and has a simple structure and easy operation, which will provide parameter guidance for FMF designing and the FMF sensing experiment optimizing.

## 1. Introduction

In the mode-division multiplexing system, few-mode fibers (FMFs) are not only used as the transmission medium but also used in the components of the transmission system, such as the few-mode Erbium-doped fiber amplifier (FM-EDFA) [1,2] and wavelength selective switch [3]. Meanwhile, FMFs have also shown their application prospects in the sensing field [4]. Therefore, the description and testing of the fiber characteristics have become more important.

After the FMF designing and drawing, the fiber characterization is essential for optimization, which is helpful for further fabrication and adjusting parameters of the transmission system, such as mode dispersion and mode coupling value. For instance, the mode dispersion of FMFs affects the step size of the receiver’s frequency domain equalization [5], and the mode coupling value determines the complexity of the receiver’s multiple-input-multiple-output (MIMO) processing system [6]. In the optical fiber sensing field, the mode coupling value will also influence the experimental results [7]. Hence, the application will be greatly developed if the FMF parameters are measured in advance.

In mode dispersion measurement, R. Gabet et al. proposed the optical low-coherence interferometry (OLCI) method, which combined with the “time-wavelength mapping” algorithm, to measure the mode dispersion [8]. The advantage of this method that in can be performed without the aid of the mode conversion device and the measurement of each mode’s absolute mode dispersion value. However, the setup is more complicated, and the experimental process requires high precision in operation. Lixian Wang et al. proposed a microwave interference technique to measure differential mode group delay (DMGD) in FMFs [9]. The advantages of this method are the simple setup, low cost, easy access to various parts of the device, and low bandwidth requirements, and it can measure C and L segments of fiber with a long length. However, when the testing system is complicated, this method has a single measurement parameter and can only measure the DMGD value of the optical fiber and cannot measure other optical fiber parameters simultaneously.

There is no mode coupling between modes in the ideal FMF, and the information in modes is transmitted independently, as Figure 1a shows. In the real fiber, coupling between different modes or polarization states will occur with power exchanging and crosstalk due to the existence of perturbation [10], as Figure 1b shows.

Concerning types of perturbation, the first is random perturbation induced by imperfections in the fiber drawing, such as the ellipticity of the fiber core, the uneven refractive index distribution of the fiber core, and cladding, or the micro-bending of the fiber. This kind of perturbation is unavoidable and has a low magnitude [11]. Another is the planned perturbation artificially introduced to reduce DMGD and apply to long-distance mode-division multiplexing transmission systems [12]. For example, the propagation constants of different modes are made to be similar in their fiber structure design. Then, the strong coupling will generate in a short transmission distance [13]. Masataka Nakazawa et al. proposed a simultaneous multi-channel optical time-domain reflectometer (OTDR) to measure this kind of mode coupling in the FMF [14]. Although this method can accurately measure the value of mode coupling in the FMF, the system is complicated and costly.

The above methods can achieve accurate measurements of mode dispersion and mode coupling of FMFs, respectively, but they have two major drawbacks: first, the optical device is complicated and expensive, which makes it difficult to measure in a timely manner; second, the test results are relatively single and cannot acquire both of dispersion and coupling value at the same setup. As a result, it is necessary to build an FMF characterization system with a low complexity that can simultaneously measure mode dispersion and mode coupling.

The normal spatially and spectrally imaging (S^2^) technique was proposed to describe characteristics of optical fibers by Nicholson et al. [15] in 2008. The combination of the amplified-spontaneous-emission (ASE) laser source and optical spectrum analyzer (OSA) is applied in the setup and tested the large-mode-area mode through the fiber, as Figure 2a shows. The optical beam is emitted from a broadband laser source across the fiber under test (FUT), then enters the single-mode fiber (SMF) through an amplification system after a length of fiber transmission. One side of the SMF is connected to a spectrometer, and the other side is fixed on a translation stage that can scan in the x and y directions. A computer controls the movement of the translation stage, and the spectrometer can obtain the spectral information of any point (x, y) on the phase plane. This combination can obtain an accurate testing result because of the large dynamic range of the spectrometer. However, the system is complicated and time-consuming due to the single-mode fiber probes and the translation stage.

In 2009, the OFS lab updated the data processing method and the setup of the S^2^ testing technique because of the advantages of the tunable laser source (TLS) and the charge-coupled device (CCD) camera combination [16]. Different kinds of fibers were tested by the new setup, as Figure 2b shows, including the photonic bandgap fiber [17], the low-loss, hollow-core photonic bandgap fiber [18], the low-loss, low differential mode group delay fiber which supports nine LP modes [19], the few-mode fiber supports two modes and four modes [20], the two-mode fiber with low differential mode group delay, low mode coupling, low-loss characteristics [21]. In addition, Stephane Blin et al. also used the S^2^ technique that combines the TLS and the CCD camera to test the multi-mode fiber [22]. These previous works provide good scenarios for the S^2^ imaging technique but lack the consideration and analysis for mode coupling components caused by different modes crosstalk and fiber fabrication, which will not be conducive to further optimization for FMFs.

In this paper, FMFs and their characteristics are introduced, and the importance of the FMF characterization is pointed out. After discussing the advantages and disadvantages of different FMF testing methods and devices, the S^2^ imaging technique is selected. Previously, the S^2^ testing technique is utilized to measure the mode dispersion of FMFs with different lengths from our group [23]. As an extension of the previous experiment, the FMF characterization system is proposed and utilized to acquire the mode dispersion and mode coupling simultaneously. Compared with other research works mentioned above, this paper further extracts fundamental mode (FM) and high-order mode (HOM) and analyzes different mode coupling components (discrete and distributed mode coupling) based on the system. This research will provide parameter guidance for FMF designing and optimization for the FMF sensing experiment.

## 2. Methods

The S^2^ testing technique utilizes the mode dispersion existing between modes in the fiber. From the interference of the modes caused by this dispersion, the fiber characteristics are extracted, including mode dispersion and mode coupling.

In the FMF, spectral interference between modes is produced from the appearance of DMGD. The optical intensity of the fiber output side will change with the optical frequency. The optical field emitted from the fiber output side is the spatial superposition of the FM and HOMs. Therefore, parameters related to modes of the FMF can be obtained by spatially resolving the interference information of the spectrum from the superimposed optical field, which is the principle of the S^2^ imaging technique.

Inter-mode interferences, or beats between modes caused by interference, are known to impair the mode-division multiplexing system’s propagating performance in the propagating process. However, at the same time, inter-mode interference also can be utilized to detect the characteristics of FMFs. The S^2^ imaging technique is a technique for determining the parameters of the FMF that resolves the spectrum interference spatially generated by each HOM and the FM.

Taking a 2-mode fiber (TMF, a commonly used FMF) with a length of L, as an example, as Figure 3 shows, 2 modes are excited at the input side (z=0), the optical beam propagating through the optical fiber of length L at the output side (z=L). (1) and (2) are the distribution expression of the electric field at the input and output side, (3) is the distribution of the intensity field at the output side calculated from the complex conjugate of the electric field.
(1)$$E\left(x,y,t\right)=\left[A_{01}\psi_{01}\left(x,y\right)+A_{11}\psi_{11}\left(x,y\right)\right]e^{-j\omega t}$$
(2)$$E\left(x,y,t\right)=\left[A_{01}\psi_{01}\left(x,y\right)e^{i\beta_{01}L}+A_{11}\psi_{11}\left(x,y\right)e^{i\beta_{11}L}\right]e^{-j\omega t}$$
(3)$$I\left(x,y,\omega \right)=E\cdot E^{*}=I_{01}{\left|\psi_{01}\left(x,y\right)\right|}^{2}+I_{11}{\left|\psi_{11}\left(x,y\right)\right|}^{2}+2Re\left\{A_{01}\psi_{01}\left(x,y\right)A_{11}^{*}\psi_{11}^{*}\left(x,y\right)e^{i\left[\beta_{11}\left(\omega \right)-\beta_{01}\left(\omega \right)\right]L}\right\}$$

In (3), the optical intensity at any point in the output side of the fiber contains 2 modes of interference related to frequency. Taylor expansion (4) of the propagation constant β to the optical frequency ω is used to express the dispersion characteristics of optical fiber, which is the dependence of refractive index on optical frequency. In (4), $\omega_{0}$ is the central frequency, expansion factor is $\beta_{m}=\frac{\partial^{m}\beta }{\partial \omega^{m}}{|}_{\omega =\omega_{0}}$. As illustrated in (5) and (6), the first 2 coefficients have implications, indicating that the group velocity $v_{g}$ is the function of wavelength λ, referred to as group velocity dispersion (GVD). The GVD leads to delay and to broaden of wave propagation, and the DMGD per unit length and unit linewidth is described in (7) by a dispersion coefficient in (ps/(km·nm)).
(4)$$\beta \left(\omega \right)=\beta_{0}+\beta_{1}\left(\omega -\omega_{0}\right)+\frac{1}{2}\beta_{2}{\left(\omega -\omega_{0}\right)}^{2}+\ldots$$
(5)$$\beta_{1}=\frac{n_{eff}}{c}+\frac{\omega }{c}\frac{\partial_{n_{eff}}}{\partial \omega }=\frac{n_{eff}}{c}-\frac{\lambda }{c}\frac{\partial_{n_{eff}}}{\partial \lambda }=\frac{n_{g}}{c}=\frac{1}{v_{g}}$$
(6)$$\beta_{2}=\frac{\partial \beta_{1}}{\partial \omega }=\frac{\partial }{\partial \omega }\frac{n_{g}}{c}=-\frac{\lambda^{2}}{2\pi c^{2}}\frac{\partial n_{g}}{\partial \lambda }$$
(7)$$D=\frac{d}{d\lambda }\left(\frac{1}{v_{g}}\right)=-\frac{\omega }{\lambda }\beta_{2}=\frac{\lambda }{c}\frac{d^{2}n_{e}}{d\lambda^{2}}$$
(8)$$I\left(x,y,\omega_{0}+\Delta \omega \right)=I_{01}{\left|\psi_{01}\left(x,y\right)\right|}^{2}+I_{11}{\left|\psi_{11}\left(x,y\right)\right|}^{2}+2Re\left\{A_{01}\psi_{01}\left(x,y\right)A_{11}^{*}\psi_{11}^{*}\left(x,y\right)\right\}cos\left(\Delta \varphi_{11-01}\right)cos\left(\Delta \tau_{11-01}\cdot \Delta \omega \right)$$

The dispersion is mostly caused by two sources. The first is the material dispersion, while the second one is the effect on the optical waveguide, which includes inter-mode dispersion and intra-mode dispersion [24]. Without considering the waveguide dispersion and the material dispersion, according to (4) and (7), (3) can be expressed as (8), $\Delta \varphi_{11-01}$ is constant, $\Delta \tau_{11-01}$ is the mode group delay generated by 2 modes ($\;LP_{01},LP_{11}$) propagating along with a few-mode fiber of length L, which is the inter-mode dispersion.

According to (8), the optical intensity value at any point (*x*, *y*) at the fiber output side changes periodically with the frequency change. The change frequency is the value of DMGD. Therefore, the DMGD can be obtained by measuring the frequency of the output optical field’s spectrum.

As the number of modes increases, inter-mode interference becomes more complicated in the FMF, including the interference between the FM and HOMs and the interference between the HOMs, which means the frequency spectrum cannot be used to produce discrete and accurate DMGD values in this way. Therefore, in the S^2^ measurement technology, only the FM is effectively excited at the fiber input side, and the power of other HOMs is much smaller than the FM. In this case, the interference between the FM and HOMs is much more substantial than the interference between HOMs, thus realizing the accurate measurement of the DMGD value. When the FMF under test supports more than 2 modes, the principal is expressed as (9), m and n represent the mode numbers and the DMGD value $\Delta \tau_{mn}$ of the FM and each HOM obtained by doing Fourier transform for the angular frequency ω [25].
(9)$$I\left(x,y,\omega_{0}+\Delta \omega \right)=I_{01}{\left|\psi_{01}\left(x,y\right)\right|}^{2}+2\underset{mn}{\sum }Re\left\{A_{01}\psi_{01}\left(x,y\right)A_{mn}^{*}\psi_{mn}^{*}\left(x,y\right)\right\}cos\left(\Delta \varphi_{mn}\right)cos\left(\Delta \tau_{mn}\cdot \Delta \omega \right)$$

## 3. Results

### 3.1. Experiment Setup of the Few-Mode Fiber Characterization System

The S^2^ testing system now features two combinations: the broadband laser source and the spectrum analyzer, as well as the TLS and the CCD camera. In the combination of the broadband laser source and the spectrum analyzer, modes are excited by the ASE laser source at the input side. At the output side, an SMF probe connected to the spectrometer is used to scan the output optical field point by point to obtain the spectral information of the optical field. However, this method has many shortcomings: first, the single-mode probe scanning method increases the complexity of the experimental system. Second, the resolution of the spectrometer determines the maximum measurable DMGD of the system. The detectable length of the FUT is limited due to the spectrometer’s poor resolution. Besides, it takes a long time to collect data, which is not conducive to a fast measurement.

In the combination of the TLS and the CCD camera, the CCD camera is used to collect optical field images of composite modes. This experimental system is easy to operate, and the tunning interval of the TLS extends the length of the FUT. Simultaneously, by synchronizing the laser tuning wavelength with the data collected by the CCD camera, a rapid and accurate measurement is performed. This form of combination is applied in this work by combining the above factors.

The proposed few-mode fiber characterization system using S^2^ imaging technique is shown in Figure 4. The TLS is EMCORE TTX1994, the CCD camera is Chameleon—1.17.3-000 from the Point Grey Research Inc. (Richmond, BC, Canada), and the fusion splicer is FSM-100P. The collimating lens and the 3D axis stage are all from Thorlabs Inc. (Newton, NJ, USA).

In this setup, the TLS input the optical beam with a frequency by setting the tunning range and the tunning interval. The fusion splicer is used to realize the connection between the SMF and the FUT. Meanwhile, it can adjust the SMF and the FMF distance to ensure the optical beam can input in the FMF without so much loss. At the input side of the FMF, the HOM is excited by the offset while ensuring that the power of the excited HOM is much less than the FM. The collimating lens is used to collimate the Gaussian optical field emitted from the output side into the nearly parallel light. The 3D axis stage is used to adjust the position between the FMF output side and the collimate lens center in 3D dimension (x,y,z). The CCD camera is used to collect the composite optical field. The computer achieves the synchronization of the tunning optical frequency of the TLS and the CCD camera’s data collecting. During the experiment, when the tunning frequency of the TLS is adjusted, the CCD camera collects the corresponding optical field image emitted from the FUT.

Before the system tests the FMF characteristics, the system’s parameters need to be determined, including:(1)Starting frequency

The DMGD value Δτ is obtained by raising the frequency of the optical output with the frequency interval Δω, and the optical intensity at a series of optical frequencies is obtained eventually, according to (3) in Section 2. Since the 1550 nm wavelength is extensively applied in optical communication systems, the TLS’s starting frequency is set to 193.415 THz;

(2)Tunning interval

Based on the principle of mode interference, the measurable maximum DMGD relies on the tunning interval of the TLS. The smaller and more precise the tunning interval, the more accurate the DMGD obtained. The TLS utilized in this paper is EMCORE TTX1994, which has a minimum tunning interval 1GHz. As a result, the tunning interval of the TLS is set 1 GHz (0.001 THz);

(3)Ending frequency

The number of images collected by the CCD camera is set 512. If the amount of data is small, the optical intensity will decrease as the frequency oscillates, and the DMGD value will not be accurate. If the amount of data is too large, the experiment time will increase. Meanwhile, the number of images collected is set 512, which is also suitable for further process, fast Fourier transform. Since the tunning interval is set 0.001 THz and the TLS will be tuned 512 times, the ending frequency is 193.926 THz;

(4)Offset distance

The fusion splicer is employed on the FMF’s output side to bias the SMF and the FMF laterally to excite the FM and HOMs in the FMF. The offset distance is selected a few micrometers for the greatest outcomes in this work.

(5)Collimator

In the experimental setup, the output side of the FUT is fixed to 1 side of the 3D axis stage, and the collimating lens is fixed to another side, then the output side of the fiber is approximately at the focal point of the collimating lens by adjusting the positions in 3 directions of (x,y,z). In this work, 2 types of lenses with different focal lengths, 15.29 mm and 8.00 mm, are compared. The lens with an 8.00 mm focal length was chosen after analyzing the size of optical spots on the CCD camera.

### 3.2. Data Collection and Process

In the data collection process, the TLS utilized in the experiment, EMCORE, is controlled by the software “SunShell.exe”, and the optical frequency must be swept automatically by the TLS with a fixed tunning interval of 0.001 THz from the starting frequency 193.415 THz to the ending frequency 193.926 THz. In the meantime, the CCD camera will record the corresponding image under computer control as each optical frequency changes.

Then, the parameters of the CCD camera, such as the CCD camera gain (dB), the shutter time (ms), the optical output power (dBm ×100), need to adjust to suitable values for keeping the quality of collected images, which means making all pixel values of collected images are lower than 255, as Figure 5a shows. Otherwise, the image collected will be oversaturated and cannot be processed, as Figure 5b shows.

After the image collecting, data processing is required. In collected images, each pixel value range is from 0 to 255, representing the optical intensity of the original optical field. In the data processing step, the pixel value in the same location (k,l) of the image captured under the frequency of $\omega_{n}$ is obtained, as Figure 6 shows, and a dataset with 512-pixel values are collected.

In this process, TMFs of 2 different lengths (10 m and 100 m) are used in the experiment, proven in principle in Section 2. The change in mode field intensity of them are obtained, as Figure 7a,b shows, where the blue curve represents the change in optical intensity.

After data collecting, the fast Fourier transform is applied in these two datasets, and the relationship curve between beat amplitude (dB) and DMGD (ps) are obtained from them, respectively, as Figure 8a,b shows.

The DMGD values of the peak points in Figure 8a,b represent mode dispersion between the FM and HOM and between different HOMs of the FUT. In this experiment, only one peak point represents the mode dispersion between the FM and HOMs because utilized FUT is a TMF. From the peak points shown in Figure 8a,b, the DMGD of our TMF under 10 m and 100 m is 29.6929 ps and 287.1094 ps, respectively, which is a 10-fold relationship approximately. Therefore, the mode dispersion of the TMF is 2.9 ps/m, corresponding with fiber parameters.

### 3.3. Mode Coupling Measurement

#### 3.3.1. FM and HOM Extraction

Based on the experimental result in Section 3.2, the FM and HOM can be extracted by filtering. At the fiber output side, (10) express the electric mode and (11) is the optical intensity. Then, the function J(x,y,ω) is defined as (12).
(10)$$E\left(x,y,\omega \right)=E_{1}\left(x,y,\omega \right)+E_{2}\left(x,y,\omega \right)$$
(11)$$I\left(x,y,\omega \right)≃I_{1}\left(x,y,\omega \right)+2Re\left\{\sqrt{I_{1}\left(x,y,\omega \right)}E_{2}\left(x,y,\omega \right)\right\}$$
(12)$$J\left(x,y,\omega \right)=\frac{I\left(x,y,\omega \right)-I_{1}\left(x,y,\omega \right)}{2\sqrt{I_{1}\left(x,y,\omega \right)}}≃Re\left\{E_{2}\left(x,y,\omega \right)\right\}$$

$I_{1}$ is the DC component of I, and its change is substantially smaller than J(x,y,ω) when the optical frequency changes because $E_{2}$ comprises interference between the FM and other HOMs. As a result, the low-pass filter extracts the DC component of I, which is $E_{1}$, as illustrated in (13).

Meanwhile, because J(x,y,ω) comprises optical field interference of $E_{2}$, which is the content of the peak part in Figure 8a,b, the bandpass filter is used to extract the HOM content of $E_{2}$. In the bandpass filter, the center frequency is set to the DMGD value, which is the peak point in Figure 8a,b, and the bandwidth of the bandpass filter is set to a small range on both sides of the peak. Then, the intensity distribution and the multi-path interference (*MPI*) value are calculated by (14) and (15). The low-pass filter and the bandpass filter results of the 100 m FMF are shown in Figure 9a,b. The integral of curves derived from the low-pass and bandpass filter, which means their area under the curves, is calculated. Then, the results are substituted into (13), (14), (15) and obtains the value of $I_{FM}$, $\;I_{HOM}\;$, and *MPI*.
(13)$$I_{FM}\left(x,y\right)=2\int d\omega \left(J_{Lowpass_{filter}}\left(x,y,\omega \right)\right)$$
(14)$$I_{HOM}\left(x,y\right)=2\int d\omega \left(J_{Bandpass_{filter}}\left(x,y,\omega \right)\right)$$
(15)$$MPI=10log_{10}\left[\frac{\iint I_{HOM}\left(x,y\right)dxdy}{\iint I_{FM}\left(x,y\right)dxdy}\right]$$

#### 3.3.2. Discrete and Distributed Mode Coupling

In this few-mode fiber characterization system, the mode excitation is achieved by the offset between the SMF and the FMF. When the optical beam is emitted from the SMF and enters the FMF, some power will couple to the HOM. The FM and the HOM are generated at the FMF output side because of the offset between the SMF and the FMF. In this case, the power of the FM is much higher than that of the HOM.

This situation occurs at the input side of the FMF, and the input optical beam is scattered into the FM and HOM of the FMF, called discrete scattering. Distributed scattering describes the power scattering between modes in the FMF because of the imperfection of the fiber structure, the uneven distribution of the refractive index, and the non-ellipticity of the fiber. Therefore, the distributed scattering occurs in the entire optical fiber, different from the discrete scattering. The conceptual diagram of the discrete and distributed scattering of the 100 m TMF is shown in Figure 10.

In this experiment, since the FM is the main mode and occupies most of the power, the mode scattering considered is the power scattering from the FM to the HOM without considering the power conversion of the HOM to the FM. Consequently, the mode coupling measured is divided into the discrete mode coupling and the distributed mode coupling.

The difference between distributed coupling and discrete coupling is not only in the position where the mode coupling occurs but also in the magnitude of the coupling power. In the discrete coupling, the power from the FM couple to the HOM is greater than the distributed coupling. Thus, the peak point corresponds to the HOM generated by the discrete coupling, and the flat parts are the HOM caused by the distributed coupling in the Figure 11.

The power ratio of the HOM generated by the total distributed coupling to the FM (16) and (17) are also obtained based on (15) of the *MPI* calculation.
(16)$$MPI_{discrete}=10log_{10}\left[\frac{\sum_{mn}\iint I_{HOM}\left(x,y\right)dxdy}{\iint I_{FM}\left(x,y\right)dxdy}\right]$$
(17)$$MPI_{distributed}=10log_{10}[\frac{\iint I_{Fullband}\left(x,y\right)dxdy-\sum_{mn}\iint I_{HOM}\left(x,y\right)dxdy}{\iint I_{FM}\left(x,y\right)dxdy}]$$

In (16), after calculating the intensity distribution of HOMs $I_{HOM}$, the ratio of its sum to the FM intensity is the discrete mode coupling. In (17), the $I_{Fullband}$ is the total mode distribution, which is derived from the full bandpass filter. After subtracting the distribution generated by the discrete mode coupling, its ratio to the FM intensity is the value of the distributed mode coupling in the optical fiber.

In our experiment, the mode coupling value of the 100 m TMF is calculated as mentioned in the steps above. The mode coupling testing results is shown in Table 1.

## 4. Discussion

The proposed few-mode fiber characterization system can measure DMGD value and optical intensity distribution of HOMs and FM, and the power ratio of HOM to FM. Compared with other testing systems and techniques, as Table 2 shows, the system described in this paper can not only measure the mode dispersion and mode coupling at the same time by the simple device and straightforward operation, but it can also complete the fiber characterization when the fiber parameters are unknown.

In the future, based on this system, the influence of external physical factors on the mode of the FMF can be continued to explore. For example, the effect of bending [26,27] and twisting [28] on modes of the FMF, the impact of polarization and rotation on asymmetric modes [29], etc. In addition, some critical factors on the modes of FMFs can be tested in this experiment setup, such as the offset distance, temperature [30], perturbation.

The S^2^ testing technique also can be regarded as a kind of mode decomposing method [31], which can be used to measure the content of the principal mode [32]. Therefore, it also can be applied in the sensing field.

Besides, there are also some people combing the deep learning and mode decomposition, for instance, mode decomposition for the multi-mode fibers based on the convolutional neural network (CNN) [33] and deep neural network (DNN) [34]. Since the S^2^ testing technique measures the characteristics of FMFs by analyzing the optical field image, the machine learning and deep learning technology can also be combined with it, which will be a future research direction.

## 5. Conclusions

In this work, the few-mode fiber characterization system based on the spatially and spectrally (S^2^) imaging technique is proposed. The characteristics of the TMF with different lengths are tested, including the mode dispersion and the mode coupling value. Meanwhile, current FMF testing methods are summarized, and their advantages and disadvantages are analyzed.

For the fiber characterization system with the S^2^ testing technique, the combination of the TLS and the CCD camera is selected. Each part of the system is analyzed. Their parameters are ensured, including the tunning interval of the TLS, the starting, and ending frequency, etc. The system is then used to test the TMFs with 10 m and 100 m. Each optical spot image is captured at the corresponding optical frequency, and the mode dispersion is obtained through specific processing steps. Under 10 m and 100 m, the DMGD of the TMF is 29.6929 ps and 287.1094 ps, respectively, which is a 10-fold relationship approximately. Thus, the mode dispersion of the TMF is 2.9 ps/m. Then, different filters are used to process data, and 2 kinds of mode coupling of the TMF are obtained. The discrete mode coupling is −27.0056 dB, and the distributed mode coupling is −13.5627 dB.

## Figures and Tables

**Figure 1 sensors-22-01809-f001:**
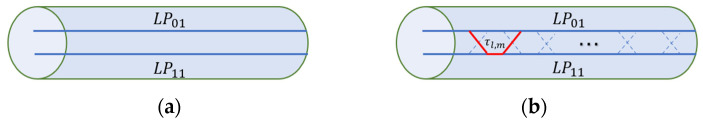
(**a**) The optical modes in ideal FMF; (**b**) the optical modes in real FMF.

**Figure 2 sensors-22-01809-f002:**
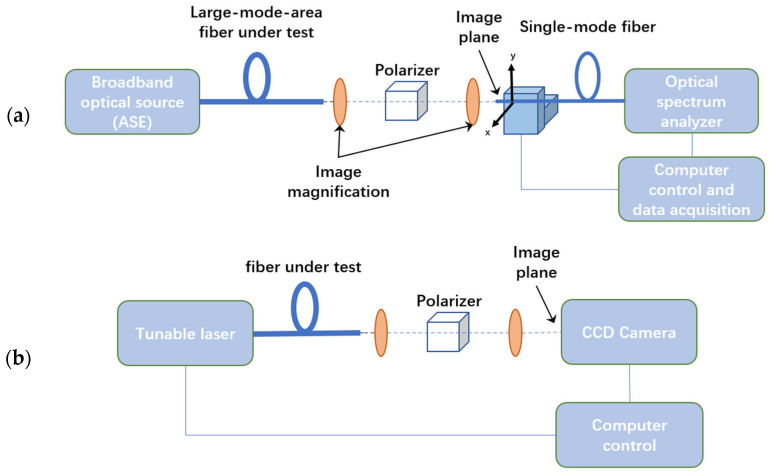
(**a**) S^2^ testing system (ASE + OSA); (**b**) Updated S^2^ testing system (TLS + CCD).

**Figure 3 sensors-22-01809-f003:**
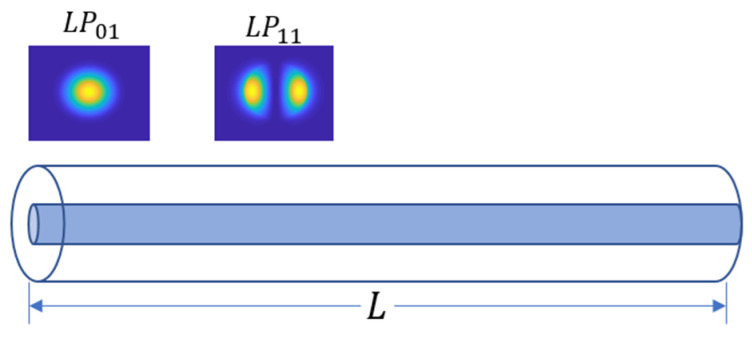
The FMF supports two modes ($\;LP_{01},LP_{11}$).

**Figure 4 sensors-22-01809-f004:**
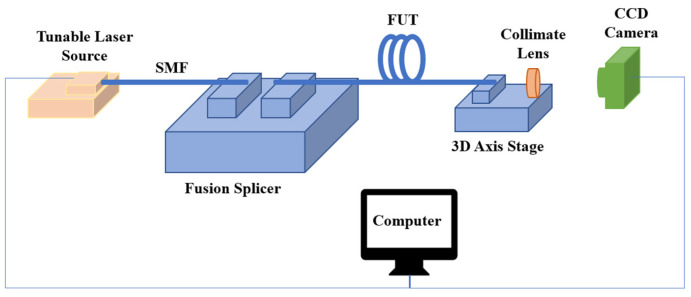
The few-mode fiber characterization system based on the S^2^ technique.

**Figure 5 sensors-22-01809-f005:**
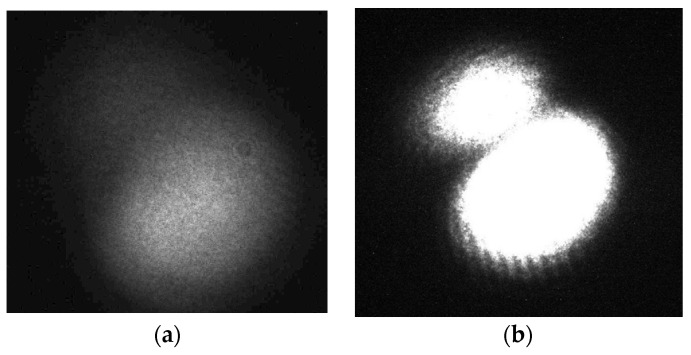
(**a**) Proper optical spot; (**b**) oversaturated optical spot.

**Figure 6 sensors-22-01809-f006:**
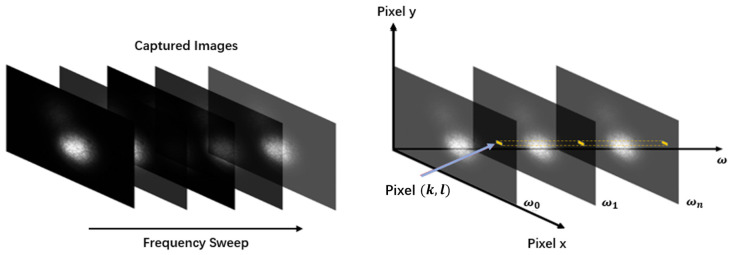
Data collecting process. Each image under different optical frequency is captured, then the pixel value in the same location (k,l) of captured images is collected.

**Figure 7 sensors-22-01809-f007:**
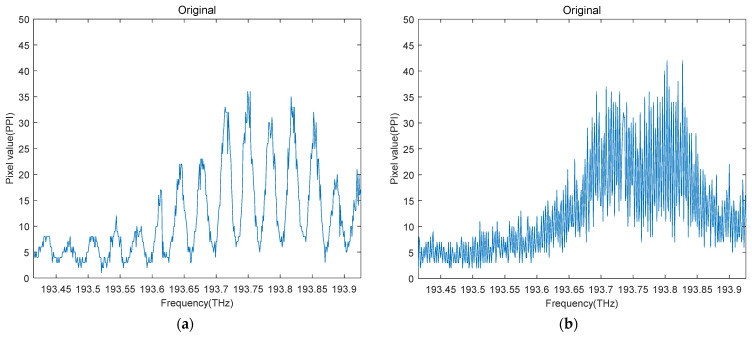
(**a**) Mode field intensity change of 10 m TMF; (**b**) mode field intensity change of 100 m TMF.

**Figure 8 sensors-22-01809-f008:**
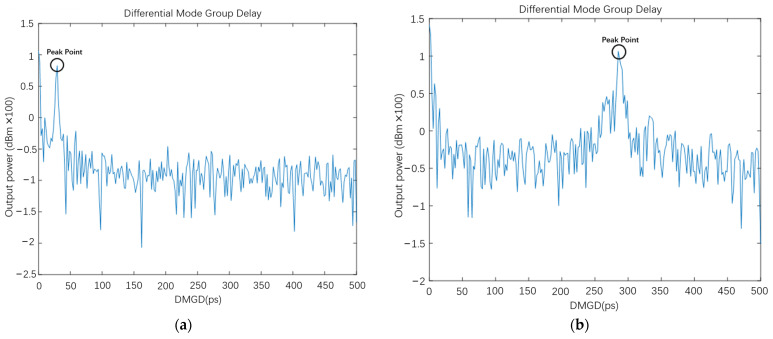
(**a**) The DMGD curve of 10 m TMF; (**b**) the DMGD curve of 100 m TMF.

**Figure 9 sensors-22-01809-f009:**
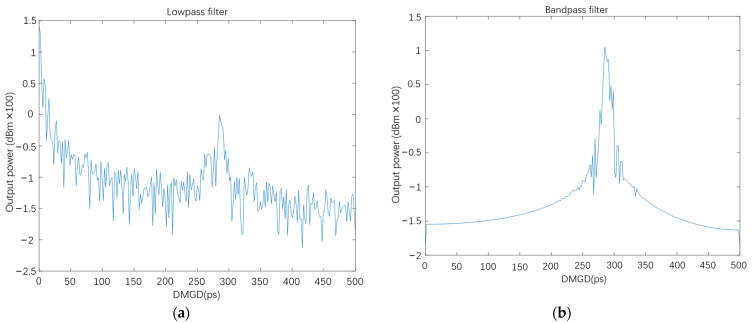
(**a**) The extracted FM; (**b**) the extracted HOM.

**Figure 10 sensors-22-01809-f010:**
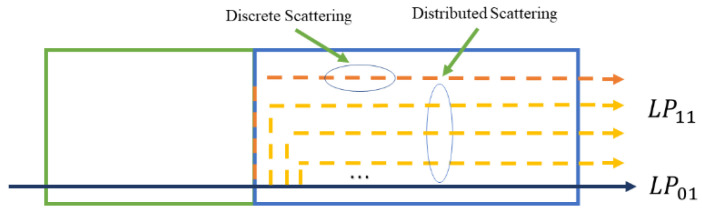
Discrete scattering and distributed scattering.

**Figure 11 sensors-22-01809-f011:**
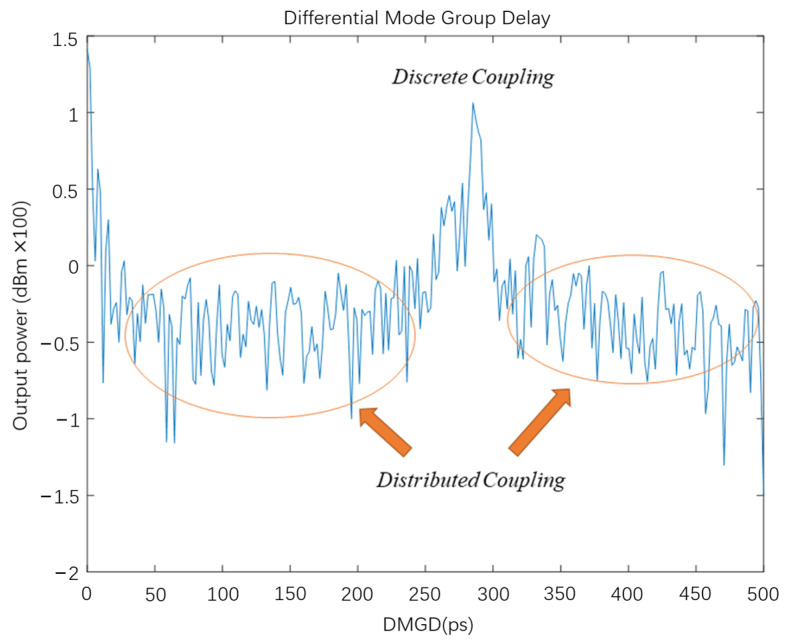
Distribution of two kinds of mode coupling in the DMGD curve.

**Table 1 sensors-22-01809-t001:** The mode coupling test result of the TMF.

Mode Coupling	Result
Discrete mode coupling	−27.0056 dB
Distributed mode coupling	−13.5627 dB

**Table 2 sensors-22-01809-t002:** Summary of optical fiber characterization techniques and advantages and disadvantages discussion.

Optical Fiber Characterization System	Measurable Fiber Properties	Advantages	Disadvantages
OLCI technique [8]	Mode dispersion	Testing without mode conversion	Complex device and difficult operation
Microwave interference technique [9]	Mode dispersion	Long FUT length; simple device; low cost	Single measurable fiber parameter
Simultaneous multichannel OTDR technique [14]	Mode coupling	Accurate test results	Complex device; high cost; Single measurable fiber parameters
The fiber characterization system in this work with S^2^ technique	Mode dispersion Mode coupling Optical intensity distribution	Testing for unkown characteristics fiber; Simple device and easy operation; productizable	Limited FUT length

## Data Availability

Not applicable.

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
