# Peer review of "Few-Mode Fiber Characterization System Based on the Spatially and Spectrally Imaging Technique†"

_sensors, 2022, doi:10.3390/s22051809_

Round 1

Reviewer 1 Report

In this paper, few-mode fibre testing techniques are discussed and their characterization system with the S2 imaging technique is built to test the mode dispersion and the mode coupling of the few-mode fibre. Thus, I recommend the following comments for revision:

  1. Authors are suggested to revise the introduction to point out more clearly what their goal was, what they achieved and what’s new about the intended results. Explore the novelty of these results and put them into a framed context. Grammatical errors and the repetition of the words (or terms) should be avoided
  2. Abstract of the manuscript should be reframed to narrate the potential outcomes of the anticipated work.
  3. It is suggested to narrate the different among the few-mode fibre (FMF) and two-mode fibre as mentioned in the manuscript at once, for the sake of readers. Both terms are used frequently but the results are tabulated for two-mode fibre with different lengths.
  4. Define the abbreviation such as for MIMO (in line 36), DMGD (in line 45), FUT (in line 80), CCD (in line 89), ASE (in line 100), MPI (in line 281) in the beginning to avoid the repetition of these terms.
  5. Avoid the use of in the end of sentences (see lines 27-28, 337-338, 339-340, 358-359)
  6. [24] is not cited in the text of manuscript.
  7. Define the abbreviation for broadband laser source (in line 80), single-mode fibre (in line 81), tunable laser source (in line 91), high-order mode (in line 108), fundamental mode (in line 117), two-mode fibre (in line 118); where they are introduced at first so that their frequent repetition can be avoided.
  8. Check the line 18-19 for grammatical error.
  9. Cite the suitable references in table 2 for the techniques tabulated under the heading of optical fibre characterization system.
  10. Elaborate the term beats of modes (line 103) or cite the appropriate references.

Reviewer 2 Report

The authors utilize S2 imaging technique to characterize two-mode fiber with different lengths. This work has merit but overall lacks novelty, thus I recommend a major revision. Comments below may be helpful for further improvements.

  1. The main method for few-mode fiber characterization in this paper is S2 imaging technique. It relies on a tunable laser source and CCD camera which is invented in reference [16], and cultivated in references [17] to [22]. In the current manuscript, the authors repeated the same method and procedures for the same purpose. Is there any distinct difference between the current work and the references? What is the novelty here, the method, the data processing or algorithm?
  2. The acronym “MIMO”, “DMGD” (line 45 and 132), ”FUT”, “OFS lab” are not explained when they show up first time in the paper.
  3. What is in figure 1(b)?
  4. In the introduction, researchers’ work in the field are introduced without necessary details. For instance, “Masataka Nakazawa et al. proposed a simultaneous multi-channel optical time-domain reflectometer (OTDR) to measure the mode coupling in the FMF [14]. Although this method can accurately measure the value of mode coupling in the FMF, the system is 68 complicated and costly. ”

Round 2

Reviewer 2 Report

The new version has addressed all my comments.